# The Outcomes of an Interventional Oral Health Program on Dental Students’ Oral Hygiene

**DOI:** 10.3390/ijerph182413242

**Published:** 2021-12-15

**Authors:** Katarina Kalevski, Jovan Vojinovic, Milica Gajic, Ema Aleksic, Zoran Tambur, Jovana Milutinovic, Nenad Borotic, Rasa Mladenovic

**Affiliations:** 1Faculty of Stomatology, Pancevo, University Business Academy in Novi Sad, 26000 Novi Sad, Serbia; katarina.kalevski@sfp.rs (K.K.); jovan.vojinovic@sfp.rs (J.V.); milica.gajic@sfp.rs (M.G.); ema.aleksic@sfp.rs (E.A.); zoran.tambur@sfp.rs (Z.T.); jovana.milutinovic@sfp.rs (J.M.); nenad.borotic@sfp.rs (N.B.); 2Department of Dentistry, Faculty of Medical Sciences, University of Kragujevac, 34000 Kragujevac, Serbia

**Keywords:** oral health, oral hygiene, dental students, health education program, intervention program

## Abstract

Dental students are an interesting population because they are expected to have a higher level of knowledge of and to exhibit better oral hygiene habits, and thus have a greater impact on the environment, families, and society as a whole. The aim of this research was to determine the state of oral hygiene in dental students before and after the interventional health education program. The research sample consisted of 119 students of dentistry in their first and fourth years of study. The first research stage was conducted before health education intervention (for the evaluation of selected oral health parameters, the Decayed, Missing and Filled index, Greene–Vermillion index, Silness–Löe plaque index, Silness–Löe gingival index, and the Community Periodontal Index of Treatment Needs index were used). The second stage was conducted as a clinical measurement of oral health changes. The conducted health education intervention, to an extent, led to changes in the state of oral hygiene among the students, and thus healthier habits. There was a statistically significant difference in the examined population between the beginning of the study and after the health education intervention program. Although a significant improvement in oral hygiene and oral health was noticed after the health education intervention program, the state of oral hygiene was still not at a satisfactory level among the dental students, contrary to our expectations.

## 1. Introduction

Dentists play an important role in improving oral health. Therefore, acquiring knowledge regarding dental health and the prevention of oral diseases is crucial during the training of dental students throughout their studies [1].

One of the main goals of health education is to train students so that they can motivate patients to adopt good oral hygiene, and there is certainly a greater possibility of them doing exactly that if they themselves are motivated and implement good oral hygiene [2]. Dental students should be able to apply their knowledge to their own dental care [3]. Their knowledge largely determines the level of maintenance of oral hygiene of their future patients [4,5]. Dental students are particularly interesting because they are expected to have a higher level of knowledge and skills, and they are also expected to exhibit better oral hygiene habits, and thus have a greater impact on the environment, families, and society as a whole. Their knowledge of oral health and how to implement oral hygiene changes and improves during their years of study [6]; the way students experience changes in their views towards oral health during their undergraduate education can be seen as a reflection of the effect of dental training. Students’ attitude towards their own teeth, proper oral hygiene, and oral health in general, can play a significant role in the later determination of health treatments [7]. 

Oral health is part of general health and affects people’s daily activities and quality of life. Activity within the field of oral health is one of the many important opportunities for preserving and improving the health and quality of life of the entire population, because it is directed towards the elimination of harmful factors that can endanger human health [8]. Health education should be distinguished from health propaganda and health edification, with which it is most often identified in everyday practice. The aim of health education is to develop responsibility for one’s own health, the health of their loved ones, and the health of those in their close proximity. Health propaganda is only one part of health education. Its aim is to inform and acquaint the public with some health issues, in order to motivate and interest them. Health edification refers to the acquisition of certain health knowledge. It must be emphasized that, unlike the other two that teach and inform, health education is effective in improving knowledge [9], which means that knowledge is received, incorporated, and applied as such. 

In dentistry, there are two basic (most common) diseases: caries and periodontitis (very often chronic diseases). Periodontal health is a major component of oral health that concentrates on the prevention of inflammatory diseases in the supportive tissue surrounding the teeth. These diseases are largely preventable and represent a serious public health problem; in most cases, the development of the disease depends on the individual and their implementation of proper and adequate oral hygiene [10,11].

There are a number of factors that greatly affect the state of oral health of students, and the younger population in general, thus affecting the individual and collective attitude towards oral hygiene and habits in maintaining it. A student’s way of life is not exclusively a matter of free choice that depends on chosen values, their attitudes and knowledge, or even set priorities, but is conditioned by environmental factors and is often limited by cultural, economic, and social factors [12]. Social determinants, such as economic factors, environmental factors, and lifestyle, have a major impact on oral health [13]. Healthy habits are activities that promote, protect, or maintain an individual’s health, while risky behaviors refer to actions that have a negative effect on oral health [14]. 

Health education is based on scientific truths about the multicausal etiology of these diseases, which results in the need to change or improve knowledge in dental students, as well as the individual, group, or community, in terms of establishing the habit of maintaining good oral hygiene (regular maintenance, techniques, means, efficiency control), the use of fluoride, and even establishing a proper diet [15].

The aim of this research was to determine the state of oral hygiene in dental students before and after an interventional health education program, to evaluate its effectiveness in order to improve the state of oral hygiene.

## 2. Materials and Methods

### 2.1. Research Method and Plan

The research was conducted as a longitudinal cross-sectional study. Research preparations and the research itself were conducted at the Faculty of Stomatology in Pancevo in two stages: the first stage was conducted before health education intervention (during September and October of the 2018 school year) conducted through a dental examination, where assessments of the level of oral hygiene were made using the appropriate index of indicators. The assessment of the oral health conditions were performed in the dental clinic of the Faculty of Stomatology in Pancevo, under artificial lightning; a dental probe, periodontal probe, and dental mirror were used in the examination. Health education intervention was conducted afterwards in all the examined students in their first and fourth years of study, with a period of six months. The second stage was conducted with all students of the research sample, after health education intervention (March 2019), as a clinical measurement of oral health changes. 

The research sample consisted of students of dentistry in their first and fourth years of study at the Faculty of Stomatology in Pancevo; there were a total of 119 students, of which 65 were first-year and 54 were fourth-year students. Students were selected as a compact group for observation, by simple random choice.

Implementation of this study was approved by the Ethics Committee of the Faculty of Stomatology in Pancevo, University Business Academy in Novi Sad, protocol number 1161/1-2017. Only those respondents who sent written consent in relation to the voluntary participation in the study were included in the study.

### 2.2. Research Instruments

The following are defined as the research instruments for this study:A research record of the oral health (oral hygiene) condition in respondents, adjusted to the WHO (World Health Organization) parameters [15]. For evaluation of selected oral health parameters, the DMF index was used to evaluate the total sum of decayed, extracted (missing), and filled teeth. For estimation of the condition of oral hygiene, the Greene–Vermillion index and Silness–Löe plaque index were used. For estimation of gingival condition, the Silness–Löe gingival index was used. For estimation of the supportive dental tissue, the CPITN (community periodontal index of treatment needs) was used.Health education intervention to improve habits in the implementation of oral hygiene [16,17]. The interventional oral health education program for the purposes of this research was defined in three stages through three complementary fields: the importance of oral health, oral hygiene, and health-safe habits. It included the method (group health education work and communication methods—live demonstrations, creative workshops), means of research (visual, audio-visual, and demonstration models), content (characteristics of good oral health, the importance of oral health for health overall, preventability of oral diseases, control and preventive examinations at the dentist, definition and explanation of terms such as dental plaque, decay, gingivitis, concretions, periodontal disease, oral hygiene, oral hygiene accessories, toothbrush technique, and individual goals in achieving good oral health), and practical work (training in proper oral hygiene, training in use of oral hygiene aids—dental floss, dental floss holder, proximal brush, mouthwash, oral and dental hygiene control—dental biofilm staining method, toothpaste selection criteria for daily use—and the interpretation of fluoride composition declaration in toothpaste).

### 2.3. Statistical Analysis

The statistical analyses were performed using the SPSS 19.0 software (SPSS Inc., Chicago, IL, USA). In order to reach relevant conclusions with respect to the surveyed groups, the obtained data for numerical characteristics are presented in the tables containing the relevant statistical parameters necessary for the statistical conclusion in the set research. The descriptive statistics methods used in the research were the arithmetic mean, standard deviation, coefficient of variation, and standard error. The methods of differential statistics used in the research were parametric tests of independent samples (confidence interval for probability *p* = 0.95, ANOVA, Levene’s test, Student’s *t*-test), a parametric test of dependent samples (paired samples *t*-test), and nonparametric tests of independent samples (Pearson Chi-Square test, Fisher’s exact tests).

## 3. Results and Discussion

This study showed that, at the beginning of the health education intervention, personal care in maintaining oral hygiene in the dental students was at a low level, and that further efforts in education are necessary to lead to a general improvement of oral health in students, as well as the application of additional hygiene methods that are effective in maintaining a good level of oral health [18]. Unsatisfactory oral hygiene among students can be explained by the fact that students had a low awareness of oral health and poor knowledge at the beginning of their dental studies. Another possible reason for this is the lack of effective school programs on the importance of oral health at the national level, which aims to help children improve their oral hygiene and maintain oral health at a desirable level [19]. The students’ oral health status at the beginning of the research showed that the average number of healthy teeth was 20.5, the number of decayed teeth for the entire sample was 0.58, the number of extracted teeth was 0.84, and the average number of filled teeth was 5.94. There was no significant difference between the first- and the fourth-year students. The average DMF index in the study group was 7.36 (Table 1, Table 2, Table 3 and Table 4). Early tooth loss and the loss of occlusal support may cause impairment of masticatory performances and changes in the neuromuscular pattern of jaw masticatory activity [20].

The higher prevalence of caries is associated with the lack of implementation of preventive measures and organized health education programs to promote health, especially in Eastern European countries [21]. That 64.6% of first-year students and 55.6% of fourth-year students had plaque on their teeth, and 70.8% of first-year students and 79.6% of fourth-year students had tartar-like plaque, indicates an inadequate control of dental biofilm, despite our expectations that that percentage should be significantly lower, especially since the study was done on dental students. It was found that 51.4% of first-year students and 48.6% of fourth-year students had changes in the gingiva requiring dental treatment, and in relation to that, 56.9% of first-year students and 46.3% of fourth-year students had changes in the periodontium. These results indicate that an integrated approach to the promotion of oral health should include all risk factors for the development of chronic diseases and that it should be raised to a higher level through the health education program. An important task that is to be achieved by health education intervention is to instill healthy oral habits in students for the prevention of oral diseases, and the first step is to provide relevant knowledge [22]. Students most often underestimate their susceptibility to caries and periodontitis, and they do not consider them as serious health problems compared to some other chronic diseases [23,24]. In order to improve oral hygiene, it is necessary to use chlorhexidine solution in concentrations from 0.12% to 0.20% without alcohol and ADS. This have been proven to have an antiseptic effect and reduce gingival inflammation, and does not cause discoloration, taste disturbance, or dry mucous membranes [25]. Concentrations of 0.20% should be used in more pronounced acute inflammatory processes on the gingiva. Poor periodontal condition was also shown by Japanese studies, where students needed dental treatment [26], while a large number of Finnish students have a better approach to oral health, which could be explained by a better approach to organized dental care [7]. Several studies have confirmed that knowledge about oral health becomes more positive with age and level of education [27,28,29]. Since the program lasted six months, the number of healthy teeth did not change, but the number of carious teeth decreased (from 0.58 at the beginning of the study to 0.42 after the targeted intervention). As a result, the number of filled teeth increased (from 5.94 to 6.03). The total DMF index decreased. This is explained by the better motivation of students to take care of their teeth after the positive effect of the health education intervention [30,31]. Also, there were changes in the state of oral hygiene. The conducted health education intervention has to some extent led to changes in the state of oral hygiene among the students, and thus has led to healthier habits and proper and regular oral hygiene in order to improve oral health. After this program, there was a decrease in soft deposits, with 86.1% of first-year students and 74.0% of fourth-year students not having soft deposits on their teeth. There was a statistically significant difference in the examined population of students between the beginning of the study and after the health education intervention program: χ^2^ = 10.846 at the level of *p* < 0.001 (Table 5). The change in the state of oral hygiene was also expressed by the changes in the values of dental concretions in the students of the examined sample. Unlike before, after the health education intervention, only 26.2% of first- and 33.3% of fourth-year students had present calculus, so there was a statistically significant difference in the examined population of students between the beginning of the study and after the health intervention program: χ^2^ = 12.829 at the level of *p* < 0.001 (Table 6). It was determined that the health education intervention also influenced the change of the condition of the gingiva and periodontium. Changes in the gingiva after the intervention were at 49.2% in first-year students and 17.0% in fourth-year students. There was a statistically significant difference between the beginning of the study and after the education: χ^2^ = 9.135 at the level of *p* < 0.001 (Table 7). The measured presence of changes on the periodontium after the intervention was 38.5% in first-year students and 46.3% in fourth-year students. There was a statistically significant difference in the examined population of students between the beginning of the study and after the intervention: χ^2^ = 10.599 at the level of *p* < 0.001 (Table 8).

It was found that the soft and hard deposits on the teeth decreased, and the intervention affected the change in the condition of the gingiva and the supporting apparatus of the teeth [32,33]. Similar research conducted at different faculties in different environments proved that the constant improvement and adoption of knowledge at the professional level is reflected in students’ oral hygiene [2,34,35,36]. When it comes to dental plaque, the situation is different. The fact that the condition of plaque after the intervention remained at the same level as before the intervention suggests that, in order to notice a change in this segment of oral health it is, on the one hand, necessary to extend the duration of health education, and on the other hand, to intensify the parts that refer to procedures and techniques for its elimination. Similar studies conducted in Kuwait, Turkey, and Croatia show how students have progressed through various health education programs and how their oral hygiene as well as knowledge of oral health has improved [19,33,37].

## 4. Conclusions

Although a significant improvement in oral hygiene and oral health was noticed after the health education intervention program, the state of the dental students’ oral hygiene was still not at a satisfactory level, contrary to our expectations, since they chose dentistry as a professional field. This statement is supported by the noticeable inflammation of the gingiva and periodontium of the students. Since dental students are considered role models for their families, friends, and patients, it is imperative to teach them the necessary skills to achieve better oral health, which can be done through continuous health education programs in addition to basic studies. Additional education on the prevention of oral diseases is needed, but such programs should start in the early years of study. The most common oral diseases, caries, gingivitis, and periodontitis, which can be prevented with a sufficient level of knowledge, can be reduced by intensive campaigns in the promotion of oral health, as well as by providing regular dental examinations and treatments by educators.

## Figures and Tables

**Table 1 ijerph-18-13242-t001:** Condition of hard dental tissues/DMF.

Statistical Parameters	Before the Health Education Intervention			After the Health Education Intervention		
	Year of Studying		Total Number	Year of Studying		Total Number
	First	Fourth		First	Fourth	
N	65	54	119	65	54	119
Minimum	0	0	0	0	0	0
Maximum	15	16	16	15	16	16
Mean	7.569	7.111	7.361	7.277	7.111	7.202
S.E. Mean	0.583	0.561	0.407	0.580	0.561	0.405
Std. Deviation	4.704	4.119	4.435	4.679	4.119	4.416
Significance of differences in average values						
*t*-test (independent samples)	*p* = 0.559			*p* = 0.203		

**Table 2 ijerph-18-13242-t002:** Decayed teeth.

Statistical Parameters	Before the Health Education Intervention			After the Health Education Intervention		
	Year of Studying		Total Number	Year of Studying		Total Number
	First	Fourth		First	Fourth	
N	65	54	119	65	54	119
Minimum	0	0	0	0	0	0
Maximum	4	3	4	3	2	3
Mean	0.692	0.444	0.580	0.569	0.241	0.420
S.E. Mean	0.122	0.114	0.085	0.116	0.083	0.075
Std. Deviation	0.983	0.839	0.925	0.935	0.612	0.818
Significance of differences in average values						
*t*-test (independent samples)	*p* = 1.484			*p* = 2.216		

**Table 3 ijerph-18-13242-t003:** Extracted teeth.

Statistical Parameters	Before the Health Education Intervention			After the Health Education Intervention		
	Year of Studying		Total Number	Year of Studying		Total Number
	First	Fourth		First	Fourth	
N	65	54	119	65	54	119
Minimum	0	0	0	0	0	0
Maximum	4	8	8	4	8	8
Mean	0.696	0.685	0.840	0.696	0.685	0.840
S.E. Mean	0.183	0.199	0.135	0.183	0.199	0.135
Std. Deviation	1.479	1.464	1.473	1.479	1.464	1.473
Significance of differences in average values						
*t*-test (independent samples)	*p* = 1.048			*p* = 1.048		

**Table 4 ijerph-18-13242-t004:** Filled teeth.

Statistical Parameters	Before the Health Education Intervention			After the Health Education Intervention		
	Year of Studying		Total Number	Year of Studying		Total Number
	First	Fourth		First	Fourth	
N	65	54	119	65	54	119
Minimum	0	0	0	0	0	0
Maximum	13	14	14	13	14	14
Mean	5.908	5.981	5.941	5.908	6.185	6.034
S.E. Mean	0.472	0.476	0.335	0.481	0.482	0.341
Std. Deviation	3.803	3.499	3.653	3.880	3.545	3.719
Significance of differences in average values						
*t*-test (independent samples)	*p* = 0.109			*p* = 0.404		

**Table 5 ijerph-18-13242-t005:** State of oral hygiene.

			Before		After	
			Year of Studying		Year of Studying	
			First	Fourth	First	Fourth
Greene–Vermillion index of soft deposits	No Soft Deposits	Number	23	24	56	40
		%	35.4%	44.4%	86.1%	74.0%
	Soft Deposits Present	Number	42	30	9	14
		%	64.6%	55.6%	13.9%	26.0%
Total		Number	65	54	65	54
		%	100.0%	100.0%	100.0%	100.0%
			*χ^2^*= 1.133 *p* > 0.05		*χ^2^* = 10.846 *p* < 0.001 *	
Greene–Vermillion index of solid deposits	No Tartar	Number	19	11	48	36
		%	29.2%	20.4%	73.8%	66.7%
	Tartar Present	Number	46	43	17	18
		%	70.8%	79.6%	26.2%	33.3%
Total		Number	65	54	65	54
		%	100.0%	100.0%	100.0%	100.0%
			*χ^2^* = 1.285 *p* > 0.05		*χ^2^* = 12.829 *p* < 0.001 *	

* statistically significant.

**Table 6 ijerph-18-13242-t006:** State of oral hygiene.

			Before		After	
			Year of Studying		Year of Studying	
			First	Fourth	First	Fourth
Silness–Löe plaque index	No Dental Plaque	number	23	25	36	30
		%	35.4%	46.3%	55.4%	55.5%
	Dental Plaque Present	Number	42	29	29	24
		%	64.6%	53.7%	44.6%	44.5%
Total		Number	64	54	64	54
		%	100.0%	100.0%	100.0%	100.0%
			*χ^2^* = 1.136 *p* > 0.05		*χ^2^* = 0.011 *p* > 0.05	

**Table 7 ijerph-18-13242-t007:** State of the gingiva.

			Before		After	
			Year of Studying		Year of Studying	
			First	Fourth	First	Fourth
Silness–Löe gingival index	Healthy Gingiva	Number	28	19	33	44
		%	43.1%	35.2%	50.8%	83.0%
	Gingiva that Requires Treatment	Number	37	35	32	9
		%	51.4%	48.6%	49.2%	17.0%
Total		Number	65	54	65	53*
		%	100.0%	100.0%	100.0%	100.0%
			*χ^2^*= 0.075 *p* > 0.05		*χ^2^* = 9.135 *p* < 0.001 *	

* statistically significant.

**Table 8 ijerph-18-13242-t008:** State of the periodontium.

			Before		After	
			Year of Studying		Year of Studying	
			First	Fourth	First	Fourth
CPITN index	Healthy periodontium	Number	28	29	40	29
		%	43.1%	50.9%	61.5%	53.7%
	Periodontium that requires treatment	Number	37	25	25	25
		%	56.9%	46.3%	38.5%	46.3%
Total		Number	65	54	65	54
		%	100.0%	100.0%	100.0%	100.0%
			*χ^2^*= 0.085 *p* > 0.05		*χ^2^* =10.599 *p* < 0.001 *	

* The total number is different because not all students were examined.

## Data Availability

Data sharing does not apply to this article as no datasets were generated during the current study.

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
