# Peer review of "The Outcomes of an Interventional Oral Health Program on Dental Students’ Oral Hygiene"

_ijerph, 2021, doi:10.3390/ijerph182413242_

Round 1
Reviewer 1 Report
This is a paper presenting the differences in oral hygiene and caries between first and four grade dental students after the implementation of a health education program. The information is very interesting because presents information from a population not usually involved in health education programs.
However, the paper is not sufficiently descriptive in the methods and in the analysis and has several inconsistencies.
Comments
Introduction.
- The paper does not present results from similar studies.
- The objective indicates that the aim es to evaluate the effectiveness of a health education program to correct and improve oral hygiene and “overall oral health” however, overall oral health is not defined or measured in the study. This has to be very clear.
Methods
- The type of study is incorrect, it is not a longitudinal cross-sectional study, this is not an observational study.
- The independent and dependent study variables are not identified nor defined.
- The health education program is not clearly described
- It would be advisable to include a hypothesis
- The study population is not defined, it is mentioned that students were selected by simple random choice, but there is no sample size specified.
- There is no specification of the duration of the intervention.
- The statistical analysis is not clear, there is no identification of the variables to be compared, many statistical tests are mentioned, but not all were presented in the results.
Results
- The description of the results is confusing, it seems that the comparisons presented are those observed at baseline between the two groups (first- and fourth-year students) and 6 months later, but not before-after comparisons by group.
- Verify the writing of the p values in the results, the signs are incorrect (><)
- were comparisons before -after for each school level group made?
- were the changes at baseline and after the intervention analyzed?
- It seems to be not useful to compare DMFT scores at baseline and 6 months later, it is very difficult to identify differences in such a short time.
- are the authors comparing the dmf t and its components according to the school level?
- in table 5, authors compare school groups at baseline and 6 months later, they are not comparing if there was a change after the intervention, just the differences between the groups in specific moments.
- commas and decimal points are noy uniform are presented to indicate decimals
- the statistical test (X2) does not seem to compare baseline and 6 months measurements, it is not the appropriate test, it compares the oral health condition between the groups
Discussion
- What is the KEP Index? It has no interpretation
- Discussion is not consistent with the objectives and results.
The conclusion is not related to the objective of the study. Authors do not conclude specifically about the effectiveness of the oral health education program.
Author Response
Esteemed Reviewer,
Thank you for your opinion and advices.
Point 1 and 2:
Introduction.
- The paper does not present results from similar studies.
- The objective indicates that the aim es to evaluate the effectiveness of a health education program to correct and improve oral hygiene and “overall oral health” however, overall oral health is not defined or measured in the study. This has to be very clear.
Response 1: Unfortunately, there is only a limited number of studies on this topic, concerning dental students. Some of the most similar ones are cited here.
Response 2: Oral health as a term was explained in only one sentence because it was not the main point of this paper. The accent was put on health education program, which is one of the main segments in promoting oral health. In order to make the goal of the work clearer (since the entire oral health is not measured of defined), we deleted that part from the aim itself.
Point 3,4,5,6,7,8,9:
Methods
- The type of study is incorrect, it is not a longitudinal cross-sectional study, this is not an observational study.
- The independent and dependent study variables are not identified nor defined.
- The health education program is not clearly described
- It would be advisable to include a hypothesis
- The study population is not defined, it is mentioned that students were selected by simple random choice, but there is no sample size specified.
- There is no specification of the duration of the intervention.
- The statistical analysis is not clear, there is no identification of the variables to be compared, many statistical tests are mentioned, but not all were presented in the results.
Response: The effort was made to explain the meaning of health education program more clearly. In the method, chapter 2.2 contains a more detailed description of what the health education intervention meant. Unfortunately we are not able to correct all other suggestions regarding methodology and statistics in this paper, but we will consider and in some future research try to be clearer.
Point 10, 11, 12, 13, 14, 15, 16, 17, 18
Results
- The description of the results is confusing, it seems that the comparisons presented are those observed at baseline between the two groups (first- and fourth-year students) and 6 months later, but not before-after comparisons by group.
- Verify the writing of the p values in the results, the signs are incorrect (><)
- were comparisons before -after for each school level group made?
- were the changes at baseline and after the intervention analyzed?
- It seems to be not useful to compare DMFT scores at baseline and 6 months later, it is very difficult to identify differences in such a short time.
- are the authors comparing the dmf t and its components according to the school level?
- in table 5, authors compare school groups at baseline and 6 months later, they are not comparing if there was a change after the intervention, just the differences between the groups in specific moments.
- commas and decimal points are noy uniform are presented to indicate decimals
- the statistical test (X2) does not seem to compare baseline and 6 months measurements, it is not the appropriate test, it compares the oral health condition between the groups
Response 10, 12, 13, 15, 16, 18: The state of oral hygiene between first and fourth year students was being compared, so that it could show us the difference in their oral hygiene before and after the program.
Response 11: We changed the labels for p values, if we understood correctly
Response 14: We very much agree that the time period was short, and in order to make final conclusions it needs to be prolonged, which we hope to achieve in the future.
Response 17: The changes were made both in tables and in the text.
Point 19,20:
Discussion
- What is the KEP Index? It has no interpretation
- Discussion is not consistent with the objectives and results.
The conclusion is not related to the objective of the study. Authors do not conclude specifically about the effectiveness of the oral health education program.
Response 19: In discussion section, the meaning of KEP index was revised, thank you for pointing it out.
Response 20: Because of the lack of papers and studies concerning this subject, we did our best to compare the results with similar studies.
Best regards,
Authors
Reviewer 2 Report
2.2. Research Instruments
- The abbreviation should be explained when it is first used.
To indicate the monitored indicators in section 2.2./1. different terms were used than in the results section, which can be confusing for the layman.
- Health education intervention was divided into four categories – method, means of research, content, and practical work. According to my opinion it is not very clear what authors wanted to say, or what exactly it was about.
Missing label 3.2. for results after the education intervention in the results section, which makes it difficult to find your way around the results. It would also help to make this section clearer, broken down into paragraphs representing the individual indicators monitored.
- Discussion
The sentence:
“The state of oral health at the beginning of the study in dental students, the average number of carious teeth was 0.58 in total, and the average KEP index was 7,361.”
doesn't make much sense.
Line 225: “…they do not consider them as serious a health problem as some…“ ? (should be „…they do not consider them as a serious health problem as some…“ ???)
Here, too, I miss any assessment of the differences between 1st and 4th year students.
Because I'm not a native speaker, I don't feel qualified to judge about the English language and style. Nevertheless, I believe that some of the phrases used (Dental students are interesting because…), and in some cases the word order, should be adjusted after consultation with a native speaker.
Author Response
Esteemed Reviewer,
Thank you for your opinion and advices. Regarding your prepositions, we did all that we could. We followed your suggestions, and did the following:
Point 1:
2.2. Research Instruments
- The abbreviation should be explained when it is first used.
To indicate the monitored indicators in section 2.2./1. different terms were used than in the results section, which can be confusing for the layman.
Response 1: The effort was made to explain all the abbreviations, and to unify the whole paper regarding the used terms.
Point 2:
- Health education intervention was divided into four categories – method, means of research, content, and practical work. According to my opinion it is not very clear what authors wanted to say, or what exactly it was about.
Response 2: All the categories were explained in section 2.2 Research Instruments. If more details are needed, please feel free to make a suggestion about what is missing.
Point 3:
Missing label 3.2. for results after the education intervention in the results section, which makes it difficult to find your way around the results. It would also help to make this section clearer, broken down into paragraphs representing the individual indicators monitored.
Response 3: A suggested by one of the Reviewers, the text was removed. Tables 5 and 6 were broken down into paragraphs representing the individual indicators monitored. (We hope that we have grasped the concept of your review.)
Point 4:
- Discussion
The sentence:
“The state of oral health at the beginning of the study in dental students, the average number of carious teeth was 0.58 in total, and the average KEP index was 7,361.”
doesn't make much sense.
Line 225: “…they do not consider them as serious a health problem as some…“ ? (should be „…they do not consider them as a serious health problem as some…“ ???)
Response 4: It was accepted and changed.
Here, too, I miss any assessment of the differences between 1st and 4th year students.
Best regards,
Authors
Reviewer 3 Report
The aim of this research was to determine the state of oral hygiene in dental students before and after the interventional health education program, to evaluate its effectiveness in order to correct and improve the state of oral hygiene and overall oral health.
This observational study was well conducted and the results are quite interesting. However, some points needs to be clarified in order to improve the scientific sound of the overall study.
Ethical committee decision protocol must be declared.
English language must be improved.
Please revise the abstract in order to better clarify the aim of the study.
Line 53. Please try to change the way you describe oral and periodontal health.
The result section is too long. Please try to fix it and to be as schematic as possible when describe the results. Comments to the results need to be discussed only in the discussion section.
The first sentence in the discussion section must be "The aim of this study..". Please revise it.
Author Response
Esteemed Reviewer,
Thank you for your opinion and advices. Regarding your prepositions, we did all that we could. We followed your suggestions, and did the following:
Point 1: Ethical committee decision protocol must be declared.
Response 1: It was added.
Point 2: English language must be improved.
Response 2: The effort was made to improve English language, as suggested by Reviewers.
Point 3: Please revise the abstract in order to better clarify the aim of the study.
Response 3: Thank you for pointing out the oversight in the abstract, it was revised and the aim was added to the text.
Point 4: Line 53. Please try to change the way you describe oral and periodontal health.
Response 4: The effort was made to explain the meaning of these terms. If it is still insufficient, please feel free to make a suggestion about what is missing.
Point 5: The result section is too long. Please try to fix it and to be as schematic as possible when describe the results. Comments to the results need to be discussed only in the discussion section.
Response 5: The result section was shortened. Tables 5 and 6 were broken down into paragraphs representing the individual indicators monitored.
Point 6: The first sentence in the discussion section must be "The aim of this study..". Please revise it.
Response 6: It was revised.
Best regards,
Authors
Reviewer 4 Report
Dear Authors, congratulations on an interesting study. I have some minor suggestions, mostly linguistic, that would need improvement:
1) the title - "The outcomes of an interventional oral health program on dental students' oral hygiene" - maybe it would sound better
line 11- Dental students are an interesting population [...]
line 12 - possess and take better care - replace with : exhibit better oral hygiene habits
Please add 1 sentence regarding the aim of the study in the abstract.
28 - important - replace with: crucial
line 31 - replace "who" - "so that they can"
line 36 - a particularly
line 39 - all the while their knowledge... - it sounds a little bit unclear
line 41 - schooling -> undergraduate education
line 42 - education -> training
line 45 - start with "Activity within the field of oral health"
line 48 - it is a little bit unclear what you mean by propaganda and edification
line 71 - a reference is needed
lines 82-84 - please add some information regarding the instruments you used for the study (what explorer, lighting, conditions etc.)
line 107 - correct -> improve
line 136- average -> mean, please add SD values
Line 141 - start with "Regarding oral hygiene,"
Tables 1-3 - I would remove the Confidence Interval from the tables, to make them a little bit clearer
Line 157 - start with "Regarding the state of gingiva..."
Tables 4-6 - they are a little bit complicated, try to simplify them
line 209 - KEP index -> replace with DMFT
Please add some references to other studies in Discussion, maybe something like https://www.thieme-connect.com/products/ejournals/html/10.1055/s-0039-1697109
https://pubmed.ncbi.nlm.nih.gov/31666985/
Overal, after some minor improvements I consider the article perfectly suitable for publication.
Author Response
Esteemed Reviewer,
Thank you for your opinion and advices. Regarding your prepositions, we did all that we could. We followed your suggestions, and did the following:
Point 1 the title - "The outcomes of an interventional oral health program on dental students' oral hygiene" - maybe it would sound better
Response 1: It was done as suggested.
Point 2 line 11- Dental students are an interesting population [...]
Response 2: It was done as suggested.
Point 3 line 12 - possess and take better care - replace with : exhibit better oral hygiene habits –Response 3: It was done as suggested.
Point 4 Please add 1 sentence regarding the aim of the study in the abstract.
Response 4: Thank you for pointing out the oversight in the abstract, the aim was added to the text.
Point 5 28 - important - replace with: crucial
Response 5: It was changed.
Point 6 line 31 - replace "who" - "so that they can"
Response 6: It was changed.
Point 7 line 36 - a particularly
Response 7: It was added.
Point 8 line 39 - all the while their knowledge... - it sounds a little bit unclear
Response 8: It was changed.
Point 9 line 41 - schooling -> undergraduate education
Response 9: It was changed.
Point 10 line 42 - education -> training
Response 10: It was changed.
Point 11 line 45 - start with "Activity within the field of oral health"
Response 11: It was added.
Point 12 line 48 - it is a little bit unclear what you mean by propaganda and edification
Response 12: A few lines were added to try and explain the difference between propaganda and edification.
Point 13 line 71 - a reference is needed
Response 13: Reference was added.
Point 14 lines 82-84 - please add some information regarding the instruments you used for the study (what explorer, lighting, conditions etc.)
Response 14: It was added.
Point 15 line 107 - correct -> improve
Response 15: It was changed.
Point 16 line 136- average -> mean, please add SD values
Response 16: As suggested by one of the Reviewers, the section of the text with the results was removed, thus all that is left are the results in the Tables.
Point 17 Line 141 - start with "Regarding oral hygiene,"
Response 17: It was added.
Point 18 Tables 1-3 - I would remove the Confidence Interval from the tables, to make them a little bit clearer
Response 18: It was removed.
Point 19 Line 157 - start with "Regarding the state of gingiva..."
Response 19: It was added.
Point 20 Tables 4-6 - they are a little bit complicated, try to simplify them
Response 20: A suggested by one of the Reviewers, Tables 5 and 6 were broken down into paragraphs representing the individual indicators monitored.
Point 21 line 209 - KEP index -> replace with DMFT
Response 21: It was changed.
Point 22: Please add some references to other studies in Discussion, maybe something like https://www.thieme-connect.com/products/ejournals/html/10.1055/s-0039-1697109
https://pubmed.ncbi.nlm.nih.gov/31666985/
Response 22: Suggested references were added.
Best regards,
Authors
Round 2
Reviewer 1 Report
The authors did not agree to modify the type of study or address the other comments on the methodology. Also, there is no possibility that a study can be an intervention (dental health education program) and observational cross-sectional and longitudinal, these are opposite designs. The study has no internal validity and cannot be replicated with the information provided by the authors.
Author Response
Respected,
We revised the manuscript to the best of our ability. We were unable to respond to all of your comments, so we apologize.
Thanks for the review!